# Soluble Fiber Inulin Consumption Limits Alterations of the Gut Microbiota and Hepatic Fatty Acid Metabolism Caused by High-Fat Diet

**DOI:** 10.3390/nu13031037

**Published:** 2021-03-23

**Authors:** Mayssa Albouery, Alexis Bretin, Bénédicte Buteau, Stéphane Grégoire, Lucy Martine, Ségolène Gambert, Alain M. Bron, Niyazi Acar, Benoit Chassaing, Marie-Agnès Bringer

**Affiliations:** 1Centre des Sciences du Goût et de l’Alimentation, AgroSup Dijon, CNRS, INRAE, Université Bourgogne Franche-Comté, F-21000 Dijon, France; mayssa.albouery@gmail.com (M.A.); benedicte.buteau@inrae.fr (B.B.); stephane.gregoire@inrae.fr (S.G.); lucy.martine@inrae.fr (L.M.); segolene.gambert@chu-dijon.fr (S.G.); alain.bron@chu-dijon.fr (A.M.B.); niyazi.acar@inrae.fr (N.A.); 2Institute for Biomedical Sciences, Center for Inflammation, Immunity and Infection, Digestive Disease Research Group, Georgia State University, Atlanta, GA 30303, USA; abretin@gsu.edu; 3Laboratoire de Biochimie Médicale, Plateforme de Biologie Hospitalo-Universitaire, F-21000 Dijon, France; 4Department of Ophthalmology, University Hospital, F-21000 Dijon, France; 5Inserm U1016, Team ‘‘Mucosal Microbiota in Chronic Inflammatory Diseases’’, CNRS UMR 8104, Université de Paris, F-75014 Paris, France; benoit.chassaing@inserm.fr

**Keywords:** fiber, inulin, high-fat diet, liver, metabolic syndrome, lipid, fatty acids, microbiota

## Abstract

Diet shapes the gut microbiota which impacts hepatic lipid metabolism. Modifications in liver fat content are associated with metabolic disorders. We investigated the extent of dietary fat and fiber-induced alterations in the composition of gut microbiota and hepatic fatty acids (FAs). Mice were fed a purified low-fat diet (LFD) or high-fat diet (HFD) containing non-soluble fiber cellulose or soluble fiber inulin. HFD induced hepatic decreases in the amounts of C14:0, C16:1n-7, C18:1n-7 and increases in the amounts of C17:0, C20:0, C16:1n-9, C22:5n-3, C20:2n-6, C20:3n-6, and C22:4n-6. When incorporated in a LFD, inulin poorly affected the profile of FAs. However, when incorporated in a HFD, it (i) specifically led to an increase in the amounts of hepatic C18:0, C22:0, total polyunsaturated FAs (PUFAs), total n-6 PUFAs, C18:3n-3, and C18:2n-6, (ii) exacerbated the HFD-induced increase in the amount of C17:0, and (iii) prevented the HFD-induced increases in C16:1n-9 and C20:3n-6. Importantly, the expression/activity of some elongases and desaturases, as well as the gut microbiota composition, were impacted by the dietary fat and fiber content. To conclude, inulin modulated gut microbiota and hepatic fatty acid composition, and further investigations will determine whether a causal relationship exists between these two parameters.

## 1. Introduction

Long-term dietary habits influence the development of chronic diseases [1]. In particular, the adoption of Western dietary habits is strongly associated with the development of metabolic disorders [2,3,4,5]. This phenotype can be reproduced successfully and robustly in experimental models [6].

Although the molecular mechanisms involved are not fully understood, it is now well established that diet influences health partly through the pressure that the dietary nutrients exert on the gut microbiota. In addition to their exposure duration and to their dose in the diet, the nature of dietary macronutrients (proteins, carbohydrates, including fibers, and fat) is a factor that differentially influences the composition of the gut microbiota by selecting and supporting the growth of bacterial communities that harbor specific metabolic capacities. Studies in humans and in mice have shown that the fat and fiber content of the diet deeply influences the composition of the gut microbiota and its function, which can, in turn, have profound impact on metabolic health [7,8,9,10,11,12,13].

The influence of fat intake as well as their source on gut microbial composition has been extensively studied [14]. Consumption of high-saturated fat diet, hereafter referred to as high-fat diet (HFD), generally results in important shifts in bacterial communities at the phylum level (e.g., decrease in Bacteroidetes and increase in Firmicutes, and Proteobacteria) as well as at the genus level (e.g., decrease in *Akkermansia muciniphila*). This reshaping of the gut microbiota modifies its functions towards higher capacities to harvest and store energy from the diet, and to induce metabolic endotoxemia and low-grade inflammation [14]. These properties make HDF-associated dysbiosis a necessary and sufficient factor to induce metabolic syndrome (MetS) and the subsequent chronic diseases [14]. Lack of soluble fiber is an important factor promoting HFD-induced metabolic disorders [15].

Dietary fibers are classified into two groups depending on their water solubility: soluble (e.g., gums, fructans, and pectins) and insoluble (e.g., cellulose, hemicellulose, and lignin). Whereas insoluble fibers resist bacterial fermentation in the colon, soluble fibers can be metabolized by the gut microbiota into bioactive molecules such as short-chain fatty acids (SCFAs), which can affect host health [16]. However, the composition and the physicochemical properties of the fibers (e.g., degree of polymerization for inulin-type fructans) are factors that can modulate their fermentation kinetics and their biological effects on the host, including on the composition of the gut microbiota [17,18,19,20,21]. Consumption of dietary soluble fibers has been shown to support the promotion of health beneficial bacteria (e.g., *Bifidobacterium* spp., *Lactobacillus* spp., *Akkermansia muciniphila,* and *Faecalibacterium prausnitzii*) and the suppression of bacteria with pathogenic potential (e.g., *Escherichia coli*) [22,23]. It is also associated with a decreased incidence of several metabolic disorders including hypertension, diabetes, obesity, as well as heart disease [13,24]. In particular, intake of inulin-type fructans has been shown to protect mice against HFD-induced MetS by modifying the composition and the functions of the gut microbiota [13,25,26,27,28,29,30].

A body of knowledge has accumulated and points to the gut microbiota as a key regulator of host lipid and fatty acid metabolism. Indeed, comparisons of germ-free (GF) mice with conventional or conventionalized mice have shown that gut microbes are required for proper digestion and absorption of dietary lipids and further fat storage [31,32]. The gut microbiota also plays an active role in controlling the hepatic expression of enzymes involved in lipid/fatty acid synthesis [33,34]. Fatty acids are a source of energy and are also fundamental constituents of membrane structure and functions. Depending on their nature, they can exert a panel of biological activities (e.g., regulation of signaling pathways, activation of transcription factors, and gene expression; precursors of bioactive lipid molecules) that influence health status [35]. In this study, we explored the impact of dietary fat and fiber content, individually and in combination, on the gut microbiota and hepatic fatty acid composition. Gut microbiota composition, hepatic fatty acid content, hepatic expression of enzymes involved in fatty acid biosynthesis, and their activities were analyzed in mice fed a purified low-fat or high-fat diet that were supplemented with insoluble (cellulose) or soluble (inulin) fibers.

## 2. Materials and Methods

### 2.1. Mice and Diets

C57BL/6J male mice (The Jackson Laboratory) were housed at Georgia State University, Atlanta, GA, USA (IACUC # 18006). At 6 weeks of age, mice were fed for 11 weeks a purified low-fat diet (LFD; 10% kcal fat, Research Diets Inc., New Brunswick, NJ, USA) or a purified high-fat diet (HFD; 60% kcal fat, Research Diets Inc.). As fiber source, LFD and HFD were containing either cellulose (LFDc, #D12450J and HFDc, #D12492) or inulin (LFDi, #D13081108 and HFDi #D13081106; source of inulin: chicory (Orafti^®^HP; BENEO-Orafti, Tienen, Belgium) with an average degree of polymerization ≥ 23) (Table 1 and Table 2). The number of mice per group was *n* = 12 for LFDc, *n* = 11 for LFDi, *n* = 12 for HFDc and *n* = 12 for HFDi.

At the end of the experimental period, feces were collected for microbiota analyses. Prior to euthanasia, mice were fasted for 15 h and mice were then weighted. Blood was collected by retrobulbar venous plexus puncture and hemolysis-free serum was generated by centrifugation using serum separator tubes for glucose determination (Becton Dickinson, Franklin Lakes, NJ, USA). Euthanasia was performed by cervical dislocation. After euthanasia, epididymal fat pad weights were measured. Livers were collected for fatty acid analyses.

### 2.2. Determination of Blood Glucose Concentration

Blood glucose concentration was measured using a glucometer (Nova Max, Waltham, MA, USA).

### 2.3. Quantification of Serum Lipids and Lipoproteins

Concentrations of plasma total cholesterol, HDL-cholesterol, LDL-cholesterol, and triglycerides were determined in samples having a suitable plasma volume (*n* = 7 for LFDc, *n* = 8 for LFDi, *n* = 6 for HFDc, and *n* = 7 for HFDi) to be analyzed using a Vista analyzer (Siemens Healthcare Diagnostics, Deerfield, IL, USA).

### 2.4. Lipid Extraction and Determination of Fatty Acid Profiles in Livers

Hepatic lipids were extracted using the Folch Procedure [36] and transmethylated using boron trifluoride in methanol [37]. The FA methyl esters (FAMEs) were analyzed by gas chromatography coupled with flame ionization detection (GC-FID) as previously described [33]. The data were processed using the ChromQuest software (Thermo Fisher Scientific Inc., Illkirch, France). They were expressed as a percentage relative to total FAMEs, defined as 100%.

### 2.5. Gene Expression

Gene expression was determined as previously described [38]. The mouse-specific primers used in this study are described in Table 3. Results were normalized to the housekeeping *Hprt* gene.

### 2.6. Microbiota Analysis by 16S rRNA Gene Sequencing Using Illumina Technology

For microbiota analysis, fresh feces were collected, snap-frozen in liquid nitrogen and stored at −80 °C. 16S rRNA gene amplification and sequencing were done using the Illumina MiSeq technology as previously extensively described [39].

### 2.7. Statistical Analyses

The data are presented as mean ± standard deviation (SD) or mean ± standard deviation of the mean (SEM). Statistical and correlation analyses were performed using the GraphPad Prism software (GraphPad Software Inc., San Diego, CA, USA). Normality was tested by using D’Agostino and Pearson test and Shapiro–Wilk test. Data from the different groups were compared by using One-Way ANOVA Multiple comparison tests corrected with Bonferroni test. The *p* values of less than 0.05 were considered as statistically significant (* *p* < 0.05, ** *p* < 0.01, *** *p* < 0.001 and **** *p* < 0.0001).

## 3. Results

### 3.1. Enrichment of Diet with Inulin Impacts the Metabolic Parameters

In this study, we compared the impact of LFD and HFD supplementation with insoluble fiber (cellulose) or soluble fiber (inulin) on several metabolic parameters associated with MetS development (Figure 1).

When added to a purified low-fat diet, inulin did not alter neither the body weight of mice, their adiposity, as assessed by fat pad mass measurement, nor their blood glucose concentration, as assessed by measuring blood glucose concentration in a fasting state (Figure 1a–c). However, the level of cholesterol, and particularly that of HDL-cholesterol, was significantly (*p* < 0.0001) decreased in LFDi-fed mice compared to LFDc-fed mice (HDL-cholesterol, LFDi: 1.31 ± 0.14 mM and LFDc: 3.10 ± 0.12 mM; Figure 1d–f).

We previously reported that mice fed HFD compared to those fed LFD develop MetS features (Figure 1a–f; [38]). We showed that the body weight of mice fed HFDi was significantly (*p* < 0.001) decreased compared to those fed HFDc (HFDi: 37.12 ± 0.39 g and HFDc: 43.67 ± 0.29 g; Figure 1a). This phenotype was associated with a decrease in fat deposition in HFDi-fed mice compared to HFDc-fed mice but the difference in fat pad mass between these two groups did not reach statistical significance (Figure 1b). Moreover, inulin significantly prevented HFD-associated dysglycemia and hypercholesterolemia (Figure 1c–f). Blood glucose concentration and levels of LDL, HDL, as well as total cholesterol in HFDi-fed mice were similar to those of LFDc-fed mice (Figure 1c–f). It should be noted that we did not observe any effect of the diets on the serum triglyceride levels of mice (Figure 1g).

All of these results show that inulin has a cholesterol-lowering effect and is effective in preventing metabolic disturbances induced by HFD.

### 3.2. Inulin Supplementation Induces Specific Changes in the Gut Microbiota of Mice Fed LFD or HFD

As diet is a major modulator of the gut microbiota, we examined next the modifications in microbiota composition according to the fat and fiber content of the diet (Figure 2 and Appendix A, Figure A1). As presented in Figure 2a, beta diversity analysis importantly revealed profound alterations of microbiota composition in HFD-fed mice compared with LFD-fed mice, along the PC1 axis, both for cellulose- and inulin-supplemented diets. Furthermore, a fiber effect was observed for both LFD- and HFD-fed mice along the PC2 axis, demonstrating that both fat and fiber, to a lesser extent, impact the fecal microbiota composition. Investigation of alpha diversity demonstrated a fat effect in decreasing microbiota alpha diversity, without any beneficial effect associated with soluble fiber supplementation (Figure 2b). Taxonomic analysis demonstrated, among numerous modifications, a decrease in the abundance of *Akkermansia muciniphila* induced by HFD when compared with LFD (Figure 2c and Figure A1), as previously described [40,41], while inulin supplementation failed to prevent such loss, further demonstrating the profound impact of dietary component in regulating this microbiota member. Moreover, Linear discriminant analysis Effect Size (LEfSe) approach demonstrated that inulin supplementation had a specific impact on the intestinal microbiota based on diet fat content, with for example, an impact on *Ruminococcus gnavus* under LFD diet and an impact on the Bacteroidetes phylum under HFD diet (Figure A1). Altogether, this microbiota composition analysis demonstrated that both fat content and fiber content have profound impacts on microbiota composition.

### 3.3. Inulin Supplementation Modulates Hepatic Fatty Acid Profile

Given the roles of the gut microbiota in the regulation of the hepatic fatty acid metabolism, together with the diet-induced modifications of the gut microbiota composition, we sought to determine and compare hepatic fatty acid profiles of mice exposed to LFD and HFD supplemented with cellulose or inulin. Analyses of fatty acids were performed by GC-FID and the amount of each fatty acid was determined relative to total fatty acid content (Figure 3, Figure 4 and Figure 5).

#### 3.3.1. Saturated Fatty Acids

Regardless of the diet, no changes were observed neither in the hepatic abundance of total saturated fatty acids (SFAs; Figure 3a) nor in that of palmitic acid (C16:0), the most abundant SFA in the liver (Figure 3b). However, there were several diet-related modifications among other SFA species. Compared to LFD-fed mice, the livers of HFD-fed mice had significantly decreased amounts of C14:0, both in condition of supplementation with cellulose (LFDc vs. HFDc; *p* < 0.0001) and inulin (LFDi vs. HFDi, *p* < 0.001), and C15:0 but only in condition of inulin supplementation (LFDi vs. HFDi; *p* < 0.001; Figure 3b). In contrast, the amounts of C17:0 and C20:0 were increased in the livers of HFD-fed mice compared to those of LFD-fed mice in condition of supplementation with cellulose (LFDc vs. HFDc; C17:0, *p* < 0.0001 and C20:0, *p* < 0.05) and inulin (LFDi vs. HFDi; C17:0, *p* < 0.0001 and C20:0, *p* < 0.01). However, the amounts of C18:0 and C22:0 were increased only in condition of supplementation with inulin (LFDi vs. HFDi; C18:0, *p* < 0.0001 and C22:0, *p* < 0.01; Figure 3b).

In addition to the effect of fat, the type of fiber incorporated in the diet modulated the hepatic content in SFAs. Indeed, hepatic amounts of C15:0 and C17:0 were significantly increased in the livers of mice fed inulin-supplemented LFD (C15:0, *p* < 0.0001 and C17:0, *p* < 0.05) and HFD (*p* < 0.0001) compared to those fed with cellulose-supplemented diets (Figure 3b). Similarly, the level of C18:0 was significantly (*p* < 0.0001) increased in the livers of HFDi-fed mice compared to those of HFDc-fed mice (Figure 3b). No effect of inulin was observed on the hepatic amounts of C14:0, C20:0, and C22:0.

#### 3.3.2. Monounsaturated Fatty Acids

The fat and fiber contents of the diets also impacted the hepatic content in monounsaturated fatty acids (MUFAs; Figure 4). The livers of HFD-fed mice had significantly decreased amounts of total MUFAs in condition of supplementation with inulin when compared to LFD-fed mice (*p* < 0.001; Figure 4a). HFD was associated with significant (*p* < 0.0001) decreases in the amount of MUFAs from the omega-7 (n-7) series regardless of its fiber content (Figure 4b,d). A significant (*p* < 0.0001) increase in the hepatic amount of C16:1n-9 was also observed in mice fed HFDc compared to those fed LFDc (Figure 4e). Whereas this fiber had no effect on the hepatic amounts of MUFAs n-7 in both LFD and HFD-fed mice, inulin significantly (*p* < 0.0001) prevented the HFD-related increase in the hepatic amount of C16:1n-9 and (*p* < 0.05) decrease in the amount of hepatic C20:1n-9 in HFD-fed mice (Figure 4d,e).

#### 3.3.3. Polyunsaturated Fatty Acids

Several diet-related changes were also observed in the hepatic content in polyunsaturated fatty acids (PUFAs; Figure 5). Whereas the hepatic amount of total polyunsaturated fatty acids (PUFAs) was similar in mice fed LFDc and HFDc, it was significantly increased in the livers of mice fed HFDi compared to those fed LFDi (*p* < 0.01; Figure 5a). These results may be ensued from modifications affecting PUFAs from the omega-6 (n-6) series. Indeed, significant increases in the amounts of total n-6 PUFAs (*p* < 0.001) and particularly those of C18:2n-6 (*p* < 0.0001), which is the most abundant n-6 PUFA and is the precursor of the fatty acids from the n-6 series, and C20:2n-6 (*p* < 0.0001) were observed in the livers of HFDi-fed mice compared to LFDi-fed mice (Figure 5c,f). Amongst PUFAs from the omega-3 (n-3) series, the hepatic levels of C18:3n-3 (*p* < 0.05) and C22:5n-3 (*p* < 0.05) were also significantly increased in mice fed HFDi compared to LFDi but these changes did not lead to significant modification in the hepatic content of total n-3 PUFAs (Figure 5e). In contrast, the amount of the only PUFA from the omega-9 (n-9) series detected in the liver, C20:3n-9, was significantly (*p* < 0.0001) decreased in mice fed HFDi compared to those fed LFDi (Figure 5g). Fat-related modifications in several PUFAs were also observed in condition of diet supplementation with cellulose. We previously reported that HFD induced an increase in the hepatic amounts of C20:2n-6 and C20:3n-6 [38]. Here we showed that C22:5n-3 and C22:4n-6 amounts were also significantly increased in the liver of HFDc-fed mice compared to LFDc-fed mice (Figure 5e,f).

Except for the C20:2n-6 whose level was significantly (*p* < 0.05) increased in LFDi-fed mice compared to LFDc mice, inulin supplementation of the LFD did not modify the hepatic PUFA content (Figure 5). In contrast, the amounts of C18:3n-3 and C18:2n-6 were significantly increased in the livers of HFDi-fed mice compared to HFDc-fed mice (*p* < 0.01, Figure 5e,f). In addition, we showed that inulin supplementation of HFD enabled to prevent the increase in the amount of C20:3n-6 caused by this diet (*p* < 0.0001, Figure 5f).

Of note, the fat and fiber content of the diets did not impact the overall hepatic ratio of n-6 PUFAs to n-3 PUFAs (Figure 5d).

Altogether, these results showed that the hepatic composition in fatty acids is modulated by both the fat and the fiber content of the diet. In addition, inulin is effective in preventing several changes affecting the hepatic fatty acid composition induced by HFD consumption.

### 3.4. Fatty Acid Elongase and Desaturase Expressions in the Liver Are Modulated by the Fat and Fiber Content of the Diet

Several studies support the role of gut microbes in the regulation of the hepatic fatty acid metabolism [34,42,43]. In view of the changes in gut microbiota and the hepatic modifications in the fatty acid content that we observed in mice fed different diets, we sought to examine the expression of several enzymes involved in the desaturation and elongation of fatty acids in the liver (Figure 6).

The expression levels of the genes encoding the desaturases FADS1 (acyl-CoA (8-3)-desaturase) and FADS2 (acyl-CoA 6-desaturase) and the elongase ELOVL1 (elongation of very long chain fatty acids protein 1) remained similar in the livers of mice regardless of the diets they were exposed to (Figure 6a,b). However, we showed that the expression level of Scd1 that encodes a delta-9 desaturase and is the rate limiting enzyme that catalyzes the biosynthesis of MUFAs, was significantly (*p* < 0.05) decreased by 3.6-fold in the livers of HFDi-fed mice compared to LFDi-fed mice (Figure 6a). We previously reported that HFD induced a decrease in the expression levels of Elovl2 and Elovl5 [38]. Here, chronic exposure to HFD also induced significant decreases in the hepatic expression levels of other elongases (Figure 6b). Indeed, the mRNA levels of Elovl3 and Elovl6 where significantly decreased by 3.0-fold (*p* < 0.0001) and 1.5-fold (*p* < 0.01), respectively, in the livers of mice fed HFDc compared to those fed LFDc. Interestingly, supplementation of the HFD with inulin limited the dysregulation of the hepatic Elovl2 and Elovl5 expression (Figure 6b). Interestingly, correlation analysis revealed some positive and negative correlations between the relative abundance of microbiota members and the expression of these hepatic genes involved in lipid biosynthesis, suggesting that direct interaction between the intestinal microbiota and hepatic gene expression could exist (Appendix A, Figure A2).

These results show that the fat content of the diet modulates the saturation and desaturation program of fatty acids in the liver. Inulin was efficient in limiting some of the HFD-induced dysregulations that affected the expression of elongases.

### 3.5. The Hepatic Activity of Enzymes Involved in the Biosynthesis of Fatty Acids Is Modulated by the Fat and Fiber Content of the Diet

The fatty acid content of the liver depends on several factors that include the amounts of precursors provided by the diet, the expression of enzymes that regulate the length and the unsaturation of the fatty acids but also the degree of activity of these enzymes. Based on the amounts of precursors and specific fatty acids, we estimated, when possible, the hepatic activity of several desaturases and elongases (Figure 7). Our results showed that HFD induced a decrease of the SCD1 activity for the production of MUFAs (C16:1n-7 and C18:1n-9). However, we observed that the production of C16:1n-7 from C16:0 was impaired regardless of the fiber content of the HFD whereas the production of C18:1n-9 from C18:0 was only decreased in mice fed with the inulin-supplemented HFD (Figure 7a). The activity of FADS1 was not modulated by the diet (Figure 7b). Nevertheless, we showed that FADS2 activity was decreased in HFD-fed mice compared to LFD-fed mice when the diets were supplemented with cellulose and that addition of inulin to HFD enabled to protect against this loss of activity (Figure 7c). Regarding elongase activities, we showed that the fat or fiber content of the diet had no impact on ELOVL3 activity (Figure 7d). In addition, ELOVL2 and ELOVL5 activities were increased in the liver of HFD-fed mice compared to LFD-fed mice (Figure 7e). Supplementation of HFD with inulin prevented this effect (Figure 7e). In addition, supplementing HFD with inulin also increased ELOVL6 activity (Figure 7g).

Altogether, these results showed that the fat content of the diet modulated the hepatic activity of several enzymes involved in the fatty acid biosynthesis and that supplementation of HFD with inulin enabled to partly prevent these changes. However, inulin had no effect on the activity of these enzymes in a standard diet.

## 4. Discussion

Quality, quantity, as well as the origin of the ingested food are factors that can deeply influence human health. Among other roles, nutrition plays a significant part in the maintenance of the symbiotic interplay between gut microbes and the host, which are essential for the proper operation of the host’s biological functions. Composition of the diet, dietary patterns, as well as dietary habits are factors that shape the gut microbiota, and destabilization of the latter can affect the health status, particularly by influencing liver metabolism [1,3]. In this study, we aimed at investigating the impact of dietary fat and fiber contents on the gut microbiota composition and lipid metabolism in the liver.

Chronic exposure to HFD is known to induce an abnormal hepatic accumulation of lipids, including fatty acids [44]. These lipid dysregulations may have deleterious consequences, such as the promotion of hepatotoxicity and chronic inflammation, as well as the exacerbation of insulin resistance [45]. In this study, we have chosen to focus on the qualitative modifications that occur in the hepatic fatty acid profile after a chronic exposure to HFD. Indeed, both the quantity and the nature of the fatty acids can influence biological processes [46,47]. We found that a chronic exposure to HFD did not modify the relative abundance of total SFAs, MUFAs, and PUFAs in the liver. This result was in disagreement with previous studies showing an increase in the amount of SFAs and a decrease in that of PUFAs in the livers of HFD-fed mice compared to those of LFD-fed mice [44,48]. However, this discrepancy could ensue from variations in different parameters such as the fatty acid content of the diets, the duration of exposure to the diets, the type of targeted lipids for the fatty acid analysis, and the differences in the genetic background of the used mouse strains [49]. However, when considering the qualitative composition of hepatic lipids, several changes in the fatty acids profile were observed. Among SFAs, HFD impaired only the relative abundance of a few species that are poorly represented in the liver. These modifications concerned the C14:0, whose hepatic amount was decreased, and the C17:0 and C20:0, whose hepatic amounts were increased in HFD-fed mice. In addition, we showed that HFD feeding led to substantial modifications among hepatic MUFAs. In particular, the amount of the MUFAs from the n-7 series (C16:1n-7 and C18:1n-7), which represent 10% of total fatty acids in the liver of LFD-fed mice, was twofold decreased in the livers of HFD-fed mice. Moreover, the amount of C16:1n-9 was increased in the liver of HFD-fed mice. However, the amount of C18:1n-9, the most abundant MUFA in the liver, was unaffected by the diet. Finally, the hepatic amounts of several PUFAs (C22:5n-3, C20:2n-6, C20:3n-6 and C22:4n-6) were increased in the livers of HFD-fed mice. Although some parameters were not similar (e.g., mouse strain, and formulation of the HFD), it is interesting to note that some of these alterations (decreases in the amounts of C14:0, C16:1n-7 and increases in the amounts of C16:1n-9 and C22:4n-6) were similar to those reported by others in the liver of mice [44]. Some of the alterations we highlighted could have biological consequences. Myristic acid (C14:0) can be transferred to proteins by processes named N-myristoylation and lysine myristoylation [50]. These posttranslational modifications influence several physiological processes (e.g., immune responses, cell proliferation, differentiation, survival, and cell death) by modifying or stabilizing protein conformation, influencing protein–protein interactions, enhancing subcellular targeting endomembranes and plasma membranes, and to receptors. In addition, several immune-metabolic effects have been reported for palmitoleic acid (C16:1n-7) and vaccenic acid (C18:1n-7). In particular, they exert beneficial effects on MetS by reducing inflammation and improving some of its features such as hepatic lipid deposition and insulin sensitivity [51]. Finally, high levels of dihomo-gamma-linolenic acid (DGLA, C20:3n-6) have been reported in the plasma of obese subjects and in patients with liver steatosis [52,53].

These changes in the fatty acid content of the liver may have different origins. Indeed, we cannot exclude that they are directly related to the fatty acid formulation of the diet. Indeed, the amounts of C17:0, C20:0, C16:1, C22:5n-3, and C20:3n-6 and some precursors such as C20:4n-6 were higher in HFD compared to LFD. Another hypothesis is that the changes in the fatty acid content of the liver result from modulation of host lipid metabolism. Indeed, HFD-associated changes in the fatty acid content of the liver could also be related to modifications in the expression levels or activity of the enzymes involved in their biosynthesis. The biosynthesis of MUFAs from SFAs is catalyzed by SCD-1, also known as delta-9 desaturase. Its substrates are C16:0 and C18:0, which yield 16:1n-7 and 18:1n-9, respectively. Conflicting results have been reported in the literature regarding expression of *Scd1* in mice exposed to HFD and/or in mouse models of MetS [42,43,54]. In our study, we showed that the hepatic expression level of *Scd1* was unchanged by the diet. Moreover, when looking at the hepatic SCD-1 activity towards its different substrates, we observed that, while the conversion of C18:0 into C18:1n-9 was unchanged, the conversion C16:0 into C16:1n-7 was a 3.9-fold decrease in the liver of HFD-fed mice. Others have already described such ambivalence in the SCD1 activity regarding its substrate to produce preferentially n-7 or n-9 MUFAs in mice fed with HFD [44]. Regarding the biosynthesis of PUFAs, their degree of unsaturation and their length are regulated by desaturases in conjunction with elongases. In agreement with previous studies, the livers of HFD-fed mice had no modifications in the expression levels of *Fads1* and *Fads2* mRNAs, which encode a delta-5 desaturase and a delta-6 desaturase respectively but had a decrease in the activity of FADS2 [43,44,55]. In addition, while a decrease in the expression levels of several genes involved in the elongation of fatty acids (*Elovl2*, *Elovl3*, *Elovl5,* and *Elovl6*) was observed as already reported by others [43], the activity of ELOVL5 and ELOVL2 were increased in the liver of mice fed a HFD. Such results could explain, at least in part, the increase of the hepatic amounts that we observed for some PUFAs. Finally, in accordance with a previous study, we found that ELOVL6 activity was not modified in the liver of mice fed with a HFD [44].

Inulin is a non-digestible functional polysaccharide that belongs to the fructan carbohydrate subgroup. It is composed of β-D-fructosyl units (from 2 to 60 monomers) that are linked together by (2→1) glycosidic bonds and it usually ends with a (1→2) D-glucosyl moiety. It is present in some plants such as chicory roots, Jerusalem artichoke tubers, salsify roots, garlic bulbs and leek bulbs. The daily intake of inulin is estimated to be between 1 and 10 g in Western countries with a higher consumption by Europeans than North Americans [56,57,58,59]. So far, no toxicity has been reported for inulin consumption neither in animals nor in humans. However, gastrointestinal intolerance can be observed when inulin is consumed at high levels, with the daily dietary fiber recommendations for humans being a dose of up to 20 g. In this study, diet was supplemented with a relatively high dose of 200 g of inulin/kg of food for 11 weeks. Follow up studies appear warranted to study the impact of lower dose on a longer period in order to mimic humans’ exposure. A limitation of this study is that the food intake has not been measured so the daily dietary inulin intake cannot be estimated. 

Insoluble fibers, such as cellulose, are generally poorly fermented by gut microbes. Cellulose is used as a core constituent of HFD to prevent diarrhea. We previously reported that the amount of cellulose present in the HFD (50 g/kg compare to 200 g/kg) only modestly impact the microbiota composition compared to inulin supplementation [13]. Inulin transits along the upper gastrointestinal tract without being neither absorbed nor degraded until it reaches the colon, where it is fermented by specific bacterial groups of the microbiota that will generate SCFAs. The ability of inulin to prevent MetS through gut microbiota alterations has been convincingly shown in several experimental models [13,25,26,60]. We showed that mice fed HFDi gained less weight compared to those fed HFDc. The origin of this phenotype remains to be determined. However, in the context of our study we could not exclude that it could also be due to the difference of energetic value of the diets even if it is slight (HFDc: 5.24 kcal/g and HFDi: 4.6 kcal/g) or to a different food intake between the two groups of mice, a parameter that we have not evaluated. Indeed, it has been already reported that food intake is reduced in mice fed with compositionally defined diet supplemented with inulin [15]. However, other studies showed that there is no impact of inulin supplementation of HFD on food intake [25] while others reported that food intake in mice fed HFD plus long-chain inulin was higher than those of mice fed HFD and HFD plus short-chain inulin [19]. Several studies suggest that the anti-obesogenic effects of inulin depend on the dietary dosage or the degree of polymerization, which are factors that could influence its fermentation into SCFA by the gut microbiota [18,61,62].

In this study, we did not observe any effect of inulin supplementation to the LFD or the HFD on the amount of plasma triglycerides in a fasting state. The effect of inulin on triglyceridemia is not yet clearly established. Indeed, some discrepancies have been reported according to factors such as the fasting state of the mice at the time of the analysis, the diet in which the fiber is incorporated or its degree of polymerization [13,63,64]. Inulin consumption was also associated with a decrease in plasma cholesterol levels in mice fed LFD. Several hypotheses have been proposed to explain the cholesterol-lowering effect of inulin such as modifications of the bile acid metabolism and modulation of the hepatic lipogenesis [65].

Here, we focused on the effect of diet supplementation with inulin on liver fatty acids. We showed that supplementation of a standard mouse diet with inulin only led to slight changes in the fatty acid profile, concerning exclusively the poorly represented fatty acids (C15:0, C17:0, and C20:2n-6). In addition, in such a diet, inulin had no impact neither on the expression of desaturases and elongases at the transcript level nor on the activity of these enzymes. However, we found that supplementation of HFD with inulin led to substantial modifications of the fatty acid profile in the liver. In fact, in this diet, several effects of inulin can be distinguished. First, supplementation of HFD with inulin, as for LFD, induced an increase in the amount of C15:0. A second effect of inulin was the modification of the amounts of several fatty acids (e.g., increases in the amounts of C18:0, C22:0, total PUFA, total PUFA n-6, C18:3n-3, and C18:2n-6) in the liver when this fiber was combined to HFD. A third effect of inulin was the exacerbation of some of the modifications induced by the HFD (e.g., increase in the amount of C17:0). A last effect of inulin was a preventive one. Indeed, we showed that supplementation of HFD with inulin prevented the hepatic increases in the amounts of C16:1n-9 and C20:3n-6.

Modifications of the fatty acid profile in the liver of mice fed with diets supplemented with inulin could result from the modulation of the expression of desaturases and elongases and/or of their activity. We observed that supplementation of HFD with inulin highly reduced the amount of *Scd1* mRNA and impaired the activity of SCD1 for the production of C18:1n-9 from C18:0. Interestingly, inulin also prevented the dysregulation of ELOVL2 and ELOVL5 at both the transcript and the activity levels.

Diet is an important modulator of the composition and the function of the gut microbiota. Depending on its composition, diet exerts selective pressures on the gut microbes and allows the growth of specific bacteria species that are genetically equipped to metabolize dietary substrates or their derivative products. There is accumulating evidence pointing out to the gut microbiota as a key factor linking long-term dietary habits to host metabolic status. A role of gut microbes in the absorption and storage of lipids and in the regulation of hepatic lipogenesis has been highlighted by several studies [31,32,33,34]. Chronic consumption of a diet high in fat and low in fiber is known to promote profound changes in the composition of the gut microbiota, including in microbes identified as major triggers of metabolic dysregulations [66].

As expected, chronic exposure to a HFD led to profound alterations in microbiota composition. Interestingly, both “fat” and “fiber” effects were observed, demonstrating that both of these diet components impact the fecal microbiota composition. Among numerous microbiota alterations, taxonomic analysis demonstrated a decrease in *Akkermansia muciniphila* induced by HFD when compared with LFD, as previously described [25,40,41]. Such reduction in *A. muciniphila* abundance has also been reported in patients with metabolic disorders [67,68], while supplementation with *A. muciniphila* has been shown to improve metabolic parameters of overweight and obese patients [69]. Interestingly, our data identified that inulin supplementation was not sufficient to prevent HFD-induced loss of *A. muciniphila.* Hence, both fat content and fiber content have profound effects on microbiota composition, in a way that can impact the observed hepatic phenotypes. Among the molecules produced by gut bacteria, short-chain fatty acids (SCFAs) have been identified as factors involved in the regulation of the hepatic fatty acid metabolism and in the promotion of the beneficial effects of inulin on host metabolism and inflammation [27,34]. However, inulin does not seem to act solely by modulating SCFAs through microbiota shaping. Indeed, inulin also protects against HFD-induced MetS through the elicitation of IL-22 expression in a microbiota-dependent manner [13].

Inulin is a widely used ingredient in foods for multiple applications (as prebiotic, source of dietary fiber, sweetener, etc.). However, several studies indicate that purified inulin could also have deleterious effect on health, with the promotion of colitis as well as hepatocellular carcinoma in mice models [70,71]. Thus, the nutritional and health claims related to the use of purified inulin should be accompanied with caution on its possible harmful effects under specific contexts such as genetical susceptibility or pre-existing microbial dysbiosis.

## 5. Conclusions

A high proportion of fat in the diet modified the composition of the gut microbiota and this phenotype was associated with changes of the fatty acid profile in the liver. We showed that the modifications in liver fatty acids could result, at least in part, from dysregulations of the expression levels of enzymes involved in fatty acid elongation and desaturation and/or of their activities. When added to a standard diet, inulin poorly affects the gut microbiota composition and the hepatic profile in fatty acids. In combination with HFD, inulin prevented some of the modifications induced by the HFD but also exacerbated others. Our results show that inulin might act by partly rescuing the growth of bacterial species that are impaired in condition of HFD and by regulating the expression and the activity of desaturases and elongases. In addition to bringing new insights to the molecular mechanisms linking diet to the hepatic metabolism of fatty acids, this study sheds the light on the precautions that should be taken regarding the use of inulin for the design of dietary interventions due to the ambivalence of this fiber in preventing or exacerbating potential deleterious effect of HFD. 

## Figures and Tables

**Figure 1 nutrients-13-01037-f001:**
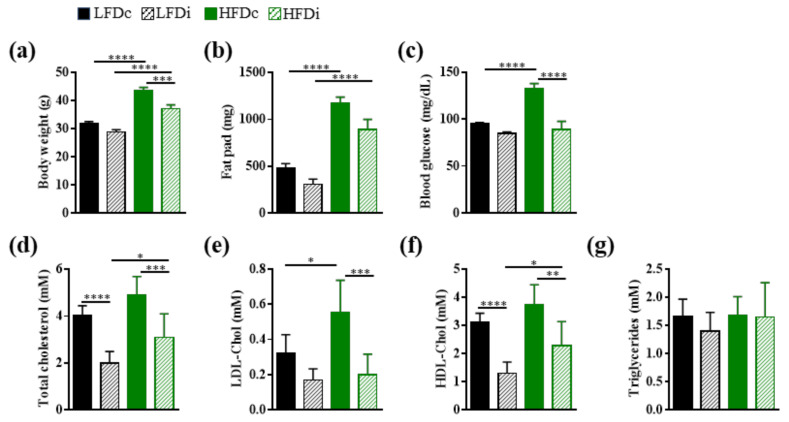
Metabolic parameters. Mice were fed low-fat diet (LFD) supplemented with cellulose (LFDc) or inulin (LFDi), or high-fat diet (HFD) supplemented with cellulose (HFDc) or inulin (HFDi) for 11 weeks. Mice were monitored for body weight (**a**), fat pad (**b**), 15h-fasting blood glucose (**c**) and serum concentration of total cholesterol (**d**), LDL-cholesterol (LDL-Chol; (**e**)), HDL-cholesterol (HDL-Chol; (**f**)) and triglycerides (**g**). Body weight, fat pad and blood glucose analyses are presented as mean ± SEM (*n* = 12 for the LFDc, HFDc and HFDi groups and *n* = 11 for the LFDi group). Lipid and lipoprotein concentrations are presented as mean ± SD (*n* = 8 for the LFDi group, *n* = 7 for the LFDc and HFDi groups, and *n* = 6 for the HFDc group). * *p* < 0.05, ** *p* < 0.01, *** *p* < 0.001 and **** *p* < 0.0001.

**Figure 2 nutrients-13-01037-f002:**
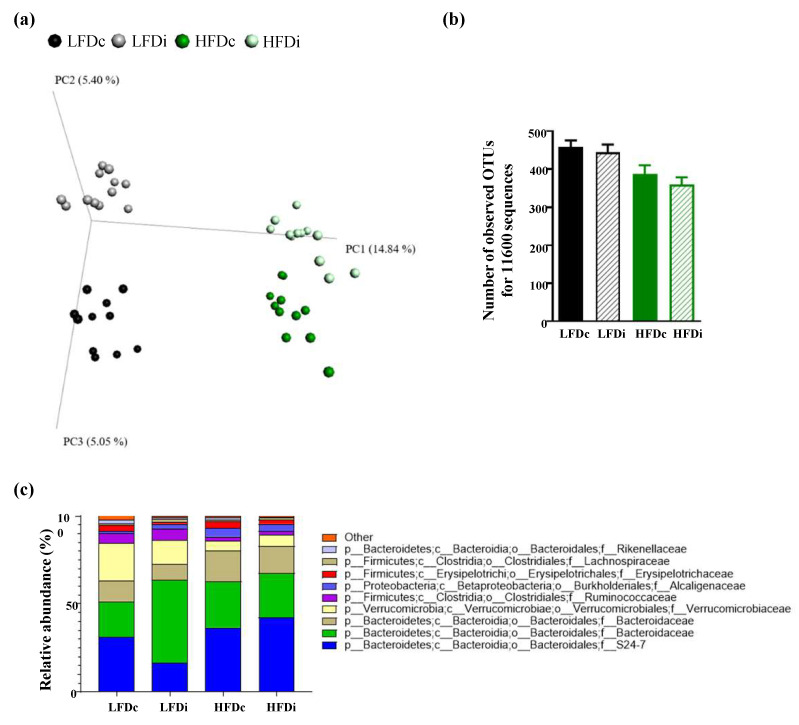
Microbiota composition analysis. Mice were fed low-fat diet (LFD) supplemented with cellulose (LFDc) or inulin (LFDi), or high-fat diet (HFD) supplemented with cellulose (HFDc) or inulin (HFDi) for 11 weeks. (**a**) Microbiota composition was analyzed at the final time point, and principal coordinate analysis of the unweighted Unifrac distance was plotted. (**b**) Alpha rarefaction was analyzed using the number of observed OTUs, and (**c**) microbiota was taxonomically summarized at the family level. For alpha diversity analyses, data are presented as mean ± SEM (*n* = 12 for the LFDc, HFDc and HFDi groups and *n* = 11 for the LFDi group).

**Figure 3 nutrients-13-01037-f003:**
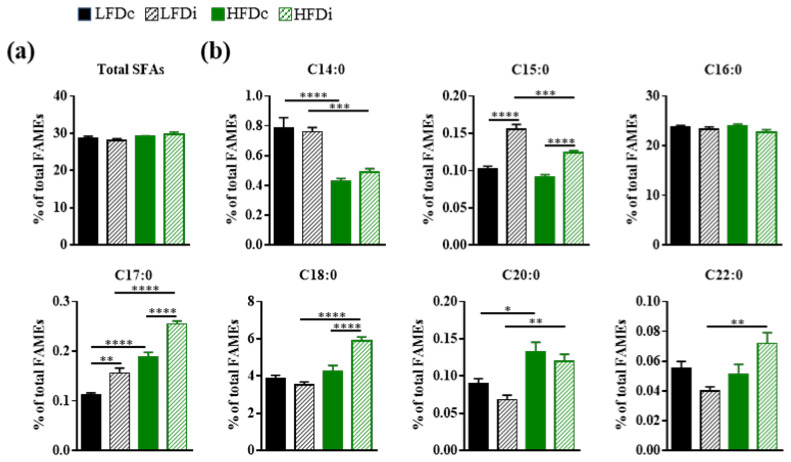
Hepatic content in saturated fatty acids. Mice were fed low-fat diet (LFD) supplemented with cellulose (LFDc) or inulin (LFDi), or high-fat diet (HFD) supplemented with cellulose (HFDc) or inulin (HFDi) for 11 weeks. Percentages of (**a**) total saturated fatty acids (SFAs) or (**b**) individual SFAs relative to total FAMEs. Data are presented as mean ± SEM (*n* = 12 for the LFDc and HFDc groups and *n* = 11 for the LFDi and HFDi groups). * *p* < 0.05, ** *p* < 0.01, *** *p* < 0.001 and **** *p* < 0.0001.

**Figure 4 nutrients-13-01037-f004:**
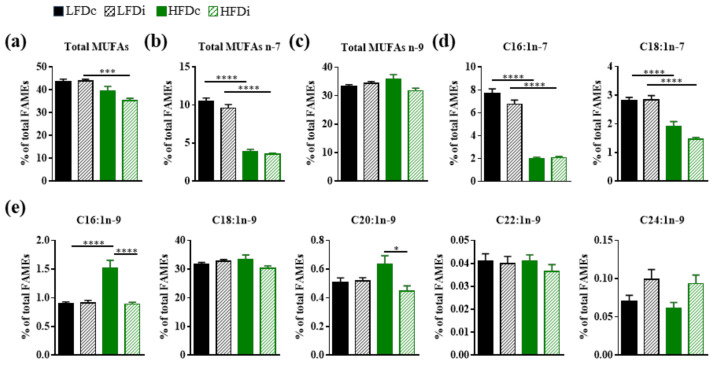
Impact of fat and inulin on the hepatic abundance of monounsaturated fatty acids. Mice were fed low-fat diet (LFD) supplemented with cellulose (LFDc, black bars) or inulin (LFDi, hatched black bars), or high-fat diet (HFD) supplemented with cellulose (HFDc, green bars) or inulin (HFDi, hatched green bars) for 11 weeks. Percentages of (**a**) total monounsaturated fatty acids (MUFAs), (**b**) total omega-7 MUFAs, (**c**) total omega-9 MUFAs, (**d**) individual omega-7 MUFAs or (**e**) individual omega-9 MUFAs relative to total FAMEs. Data are presented as mean ± SEM (*n* = 12 for the LFDc and HFDc groups and *n* = 11 for the LFDi and HFDi groups). * *p* < 0.05, *** *p* < 0.001 and **** *p* < 0.0001.

**Figure 5 nutrients-13-01037-f005:**
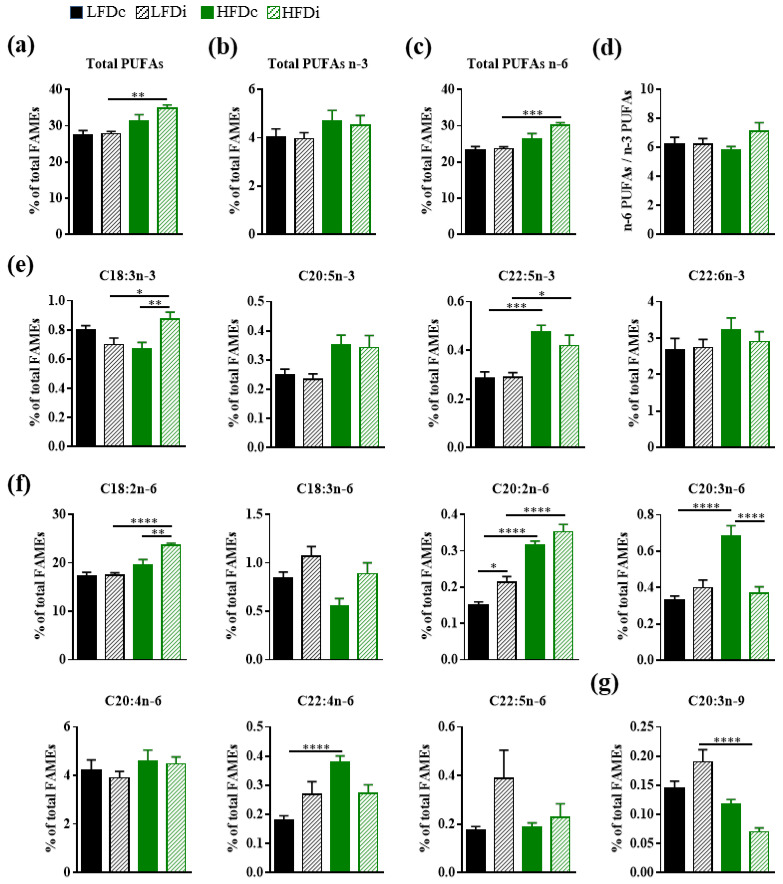
Impact of fat and inulin on the hepatic abundance of polyunsaturated fatty acids. Mice were fed low-fat diet (LFD) supplemented with cellulose (LFDc, black bars) or inulin (LFDi, hatched black bars), or high-fat diet (HFD) supplemented with cellulose (HFDc, green bars) or inulin (HFDi, hatched green bars) for 11 weeks. Percentages of (**a**) total polyunsaturated fatty acids (PUFAs), (**b**) total omega-3 (n-3) PUFAs, (**c**) total omega-6 (n-6) PUFAs, (**d**) overall n-6 PUFAs/n-3 PUFAs ratio, (**e**) individual n-3 PUFAs, (**f**) individual n-6 PUFAs or (**g**) individual omega-9 (n-9) PUFA relative to total FAMEs. Data are presented as mean ± SEM (*n* = 12 for the LFDc and HFDc groups and *n* = 11 for the LFDi and HFDi groups). * *p* < 0.05, ** *p* < 0.01, *** *p* < 0.001 and **** *p* < 0.0001.

**Figure 6 nutrients-13-01037-f006:**
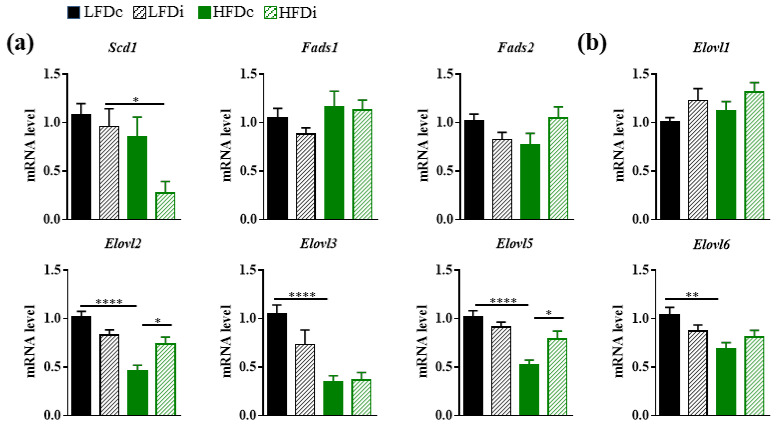
Modulation of hepatic gene expressions involved in lipid biosynthesis by the diet. Mice were fed low-fat diet (LFD) supplemented with cellulose (LFDc, black bars) or inulin (LFDi, hatched black bars), or high-fat diet (HFD) supplemented with cellulose (HFDc, green bars) or inulin (HFDi, hatched green bars) for 11 weeks. (**a**) Hepatic expression of genes encoding enzymes involved in the desaturation of fatty acids: acyl-CoA desaturase 1 (Scd1), acyl-CoA (8-3)-desaturase (Fads1) and acyl-CoA 6-desaturase (Fads2). (**b**) Hepatic expression of genes encoding enzymes involved in the elongation of fatty acids: elongation of very long chain fatty acids proteins 1, 2, 3, 5 and 6 (Elovl1, Elovl2, Elovl3, Elovl5 and Elovl6). The levels of mRNA were normalized to Hprt mRNA level for calculation of the relative levels of transcripts. mRNA levels are illustrated as fold change. Data are presented as mean ± SEM (*n* = 10-12 in each group). * *p* < 0.05, ** *p* < 0.01 and **** *p* < 0.0001.

**Figure 7 nutrients-13-01037-f007:**
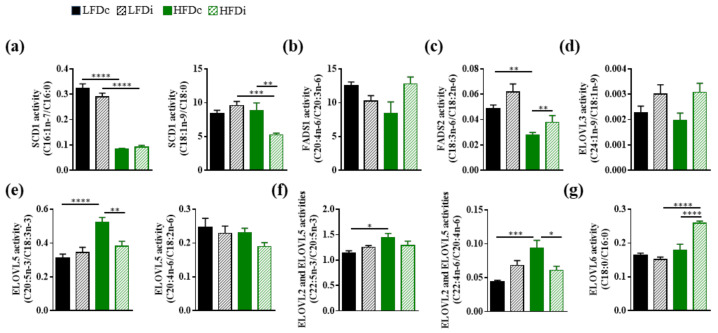
Modulation of the hepatic activities of enzymes involved in fatty acid biosynthesis by the diet. Mice were fed low-fat diet (LFD) supplemented with cellulose (LFDc, black bars) or inulin (LFDi, hatched black bars), or high-fat diet (HFD) supplemented with cellulose (HFDc, green bars) or inulin (HFDi, hatched green bars) for 11 weeks. (**a**) SCD1 activity corresponding to the ratio “product” (C16:1n-7 or C18:1n-9)/“precursor” (C16:0 or C18:0). (**b**) FADS1 activity corresponding to the ratio “product” (C20:4n-6)/“precursor” (C20:3n-6). (**c**) FADS2 activity corresponding to the ratio “product” (C18:3n-6)/“precursor” (C18:2n-6). (**d**) ELOVL3 activity corresponding to the ratio “product” (C24:1n-9)/“precursor” (C18:1n-9). (**e**) ELOVL5 activity corresponding to the ratio “product” (C20:5n-3 or C20:4n-6)/“precursor” (C18:3n-3 or C18:2n-6). (**f**) ELOVL2 and ELOVL5 activities corresponding to the ratio “product” (C22:5n-3 or C22:4n-6)/“precursor” (C20:5n-3 or C20:4n-6). (**g**) ELOVL6 activity corresponding to the ratio “product” (C18:0)/“precursor” (C16:0). Data are presented as mean ± SEM (*n* = 11–12 in each group). * *p* < 0.05, ** *p* < 0.01, *** *p* < 0.001 and **** *p* < 0.0001.

**Table 1 nutrients-13-01037-t001:** Composition of diets used in this study.

Diets	LFDc ^1^ D12450J ^5^	LFDi ^2^ D13081108 ^5^	HFDc ^3^ D12492 ^5^	HFDi ^4^ D13081106 ^5^
Protein source	Casein	Casein	Casein	Casein
Fiber source	Cellulose	Inulin	Cellulose	Inulin
Protein (kcal%)	20	20	20	20
Carbohydrates (kcal%)	70	65	20	20
Fat (kcal%)	10	10	60	60
kcal/gm	3.8	3.5	5.24	4.6
**Ingredients (g)**				
Casein	200	200	200	200
L-Cystine	3	3	3	3
Corn Starch	506.2	456.2	0	0
Maltodextrin 10	125	125	125	75
Sucrose	63.8	63.8	68.8	68.8
Cellulose	50	0	50	0
Inulin	0	200	0	200
Soybean Oil	25	25	25	25
Lard	20	20	245	245
Mineral Mix, S10026	10	10	10	10
DiCalcium Phosphate	13	13	13	13
Calcium Carbonate	5.5	5.5	5.5	5.5
Potassium Citrate, 1 H_2_O	16.5	16.5	16.5	16.5
Vitamin Mix, V10001	15	15	10	10
Choline Bitartrate	2	2	2	2

^1^ LFDc: Low-fat diet supplemented with cellulose; ^2^ LFDi: Low-fat diet supplemented with inulin; ^3^ HFDc: High-fat diet supplemented with cellulose; ^4^ HFDi: High-fat diet supplemented with inulin; ^5^ Research Diets Inc.

**Table 2 nutrients-13-01037-t002:** Fatty acid composition of the diets.

Fatty Acids ^1^	Low-Fat Diet	High-Fat Diet
C10:0	0.0	0.0
C12:0	0.0	0.1
C14:0	0.5	1.1
C15:0	0.0	0.1
C16:0	14.9	19.6
C16:1	0.7	1.3
C17:0	0.2	0.4
C18:0	7.1	10.6
C18:1	28.8	34.0
C18:2n-6	41.9	28.7
C18:3n-3	5.0	2.0
C20:0	0.0	0.2
C20:1	0.2	0.6
C20:3n-6	0.0	0.1
C20:4n-6	0.2	0.3
C20:5n-3	0.0	0.0
C22:0	0.0	0.0
C22:1	0.0	0.0
C22:4n-6	0.0	0.0
C22:5n-3	0.0	0.1
C22:5n-6	0.0	0.0
C22:6n-3	0.0	0.0
C24:0	0.0	0.0
Total SFA	22.7	32.0
Total MUFA	29.7	36.0
Total PUFA	47.1	31.2
C18:2n-6/C18:3n-3	8.3	14.1
PUFA n-6/PUFA n-3	8.4	13.7

^1^ g for 100 g.

**Table 3 nutrients-13-01037-t003:** List of primers.

Genes	Forward Primers (5′-3′)	Reverse Primers (5′-3′)
*Scd1*	CAGGAGGGCAGGTTTCCAAG	CGTTCATTTCCGGAGGGAGG
*Fads1*	CGCCAAACGCGCTACTTTAC	CCACAAAAGGATCCGTGGCA
*Fads2*	CGTGGGCAAGTTCTTGAAGC	TCTGAGAGCTTTTGCCACGG
*Elovl1*	CCTGAAGCACTTCGGATGGT	TCACTTGCCCGTCCTTCTTC
*Elovl2*	GTGATGTCCGGGTAGCCAAG	GGACGCGTGGTGATAGACAT
*Elovl3*	TACTTCTTTGGCTCTCGCCC	AGCTTACCCAGTACTCCTCCA
*Elovl5*	TGATGAACTGGGTTCCCTGC	CAGCTGCCCTTGAGTGATGT
*Elovl6*	AGAACACGTAGCGACTCCGA	TCAGATGCCGACCACCAAAG

*Scd1*: Stearoyl-CoA desaturase 1; *Fads1* and *2*: Fatty acid desaturase 1 and 2; *Elovl1*, *2*, *3*, *5,* and *6*: Elongation of very long chain fatty acids protein 1, 2, 3, 5 and 6.

## Data Availability

The data presented in this study are available on request from the corresponding author.

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
