# Peer review of "Soluble Fiber Inulin Consumption Limits Alterations of the Gut Microbiota and Hepatic Fatty Acid Metabolism Caused by High-Fat Diet"

_nutrients, 2021, doi:10.3390/nu13031037_

Round 1

Reviewer 1 Report

Author tried to answer to all the comments.

Reviewer 2 Report

The authors adequately addressed my main concern.

This manuscript is a resubmission of an earlier submission. The following is a list of the peer review reports and author responses from that submission.

Round 1

Reviewer 1 Report

The paper topic is interesting. But I would recommend authors to consider the following general and specific remarks to improve the quality of the manuscript:

The introduction was not able to really bring the problem in a general context and was not able to justify the realization of this study. The introduction is in general not convincing.

What about the role of dietary soluble fibers? What about inulin?

I could not see anywhere the nature of inulin. Besides, many plants contain only small amounts of inulin, while others are excellent sources. 

What about the black bars? Their composition? it is related to Table 1?

Table 1 is a bit confused. The ingredients presented are of black bars?

Line 238-239: no changes were observed in the hepatic abundance of total 238saturated fatty acids (SFAs; Figure 3a) nor in that of palmitic acid - correct is: neither ....nor....

Too much is used: we observed, it is your study indeed but you could present the results in a nice flowing story, not so repetitive - we observed.

Inulin poorly affects the gut microbiota composition and the hepatic profile in fatty acids - for example chicory inulin has peak fermentation 8 hours postprandially, there is no statement in this study about the consumption of dietary fiber to promote extensive metabolic interactions among bacterial species present in the gastrointestinal microbial community.

If the fiber of interest is highly fermentable, e.g., inulin-type fibers, this dosage is near the top of the tolerable limit for human consumption, and consumption at this level is likely to result in unpleasant side effects. - Please reffer to this issue, which is very important.

Reviewer 2 Report

In this paper, Albouery et al. try to explore the potential positive role of high-fat diet supplementation with inulin in an in vivo model of metabolic disease in mice. It has been widely described that intake of inulin, a dietary soluble fiber, has been shown to protect mice against HFD-induced metabolic disease by modifying gut microbiota composition and functionality. In this study, authors focused on the effect of high-fat diet supplementation with inulin on metabolic parameters, hepatic fatty acid metabolism and gut microbiota composition. Their results suggest that HFD supplementation with inulin led to substantial modification of the fatty acid profile in the liver, associated with the partial reshape of altered fatty acid metabolism-related enzymes, as desaturases and elongases. Authors suggest that the modifications in liver fatty acids could result at least in part from dysregulation of fatty acid metabolism-associated enzymes gene expression and activity.

Although the current manuscript contains some interesting aspects, it is a basic and descriptive study providing limited novel information and lacking of a mechanistic insight, and major and minor aspects should be considered:

  • Authors indicate that consumption of dietary fat and inulin determines gut microbiota composition and hepatic fatty acid profile and metabolism, concluding that inulin plays a controversial role when combined with a high-fat diet. However, they propose inconsistent conclusions about the ambivalent effects of inulin in preventing or exacerbating deleterious effect of HFD and the relationship with the partially improved gut microbiota profile and metabolic markers. The results have not been profoundly analysed. As a consequence, authors should implement improvements of results concerning gut microbiota profile and its modulation by inulin, which could be deeply described and analysed and adequately discussed.
  • Authors should consider the need of the establishment of appropriate correlations between gut microbiota composition and hepatic fatty profile and related metabolic pathways to adequately conclude this paper.
  • Discussion should be improved. It is not clearly explained if there is a good correlation between results found and the conclusions presented in this study. Sentences as “Even if some of the changes occurring in the hepatic fatty acid profile may reflect variations in the fatty acid formulation of the diets used, they could also be consequences of modifications of the gut microbiota” are not supported by the data shown, and the results presented are insufficient to confirm this sentence. An adequate study of correlations between hepatic fatty acid profile and metabolism and gut microbiota composition could be desirable, as previously indicated.
  • The impact of inulin on health are still subject of controversy. In this regard, it has been widely described its role on deleterious effects related to dysbiosis and hepatic damage exacerbation (Singh et al., Cell, 2018; Gallage et al., Cell Metab, 2018; Wen and Schwabe, Hepatology, 2019; Jia et al., Front Cell Infect Microbiol 2019; Miles et al., Inflamm Bowel Dis, 2017). Authors should place it in the context of their study.
  • Authors could consider to include relevant previous and current contributions to the background of this paper (Du et al., J Agric Food Chem, 2020; Bao et al., Front Pharmacol, 2020; Li et al., Sci Rep, 2020; Igarashi et al., PeerJ, 2020; Tan et al., Sci Rep, 2018; Weitkunat el al., J Nutr Biochem, 2015; Zhai et al., J Agric Food Chem, 2018…).
  • The methodology is incomplete. Experiments are not described in sufficient detail. In this respect, what is the average degree of polymerization of inulin used (Han et al., Mol Nutr Food Res, 2017) and its concentration on diets? How frozen feces were stored? What is the number of mice per experimental group? Differences in sample number considered in plasma metabolic determinations as lipid and lipoprotein concentrations and glucose levels could be explained. How animal sacrifice has been made?
  • Authors should adequately justify and discuss the specific effect of inulin and fat diet on Akkermansia genus abundance in gut microbiota.
  • Why faecal SCFAs has been not measured as an indicator of gut microbiota-induced inulin metabolism?
  • Could the differences observed on body weight associated to inulin supplementation be caused by distinct kcal/g between HFDc and HFDi? Has dietary intake been determined?
  • In Table 3, gene abbreviations should be added.
  • In Figure 2a, corresponding experimental groups should be indicated.
  • Figure 2c is not cited in the text.
  • Akkermansia muciniphila should be indicated in italics.
  • The language includes many errors. English should be reviewed.

Reviewer 3 Report

It is very interesting paper about regarding hepatic fatty acid metabolism and gut microbiota but it may need to be improved.

The author did not clearly write about the result. They wrote : “In conclusion, consumption of dietary fat and fiber shaped the gut microbiota and promoted changes in the hepatic FA metabolism. In combination with HFD, inulin exerted ambivalent effects”. This conclusion is too broard and like a summary of the result. No hypothesis or strategy was given.

  1. Introduction:

Main objective of the paper is: we investigated the impact of dietary fat and fiber content on the gut microbiota and hepatic fatty acid composition. (I think this objective quite old, only hepatic fatty acid composition is somehow innovative, but this has to much background noise (dietary sources, fat metabolism from other organs) if they cant reduce the background noise, their result on this is questionable)

comment.

  1. Method:

2.1. Mice and diets:

Number of mice per group?

Using insoluble cellulose as control group without the validation that there is no effect of insoluble of cellulose on microbiota composition and fat metabolism?? At least give a reference.

. comparisons between LFDc and HFDi, LFDi and HFDc seems unnecessary because there is no link between them.

No explaination for figure 2c?

Total saturated fatty acid was not affected. The paper only point out many change of FAs, some increase and some decrease without any discussion. Making the results for the whole 3 figures becomes not so useful (figure 3-5).

Figure 6

There is always the need of protein expression level. Change in mRNA expression level do not always lead to change in protein expression. And the increased activity of a gene should be explained by the level of protein instead of mRNA level

  1. Discussion:

“These changes in the fatty acid content of the liver may have different origins. They 434 could be directly related to the fatty acid formulation of the diet.”

- The author pointed out by themselves that they had a lot of background noise.

- The author list 5 effects but some are just the same (effect 2,3,4 all is about the change in fatty acid composition)

“Even if some of the changes occurring in the hepatic fatty acid profile may reflect variations in the fatty acid formulation of the diets used, they could also be consequences of modifications of the gut microbiota”

- The author pointed out by themselves that they had a lot of background noise.

“Among the molecules produced by gut bacteria, short-chain fatty acids (SCFAs) have been identified as factors involved in the regulation of the hepatic fatty acid metabolism and in the promotion of the beneficial effects of inulin on host metabolism and inflammation [16,23]”

-The author wrote and cite these references, however, they have no result related to SCFAs.

Overall for the discussion: The author mostly re-write the result, lack of discussion and explaination, no hypothesis or strategy were given.